# The 2011 Tohoku Tsunami from the Sky: A Review on the Evolution of Artificial Intelligence Methods for Damage Assessment

**Jérémie Sublime** [1,2]

1   ISEP, Institut Supérieur d'Électronique de Paris, 10 rue de Vanves, 92130 Issy-Les-Moulineaux, France; jeremie.sublime@isep.fr; Tel.: +33-1-4954-5219
2   LIPN—CNRS UMR 7030, Université Paris 13, 93430 Villetaneuse, France

**Abstract:** The Tohoku tsunami was a devastating event that struck North-East Japan in 2011 and remained in the memory of people worldwide. The amount of devastation was so great that it took years to achieve a proper assessment of the economical and structural damage, with the consequences still being felt today. However, this tsunami was also one of the first observed from the sky by modern satellites and aircrafts, thus providing a unique opportunity to exploit these data and train artificial intelligence methods that could help to better handle the aftermath of similar disasters in the future. This paper provides a review of how artificial intelligence methods applied to case studies about the Tohoku tsunami have evolved since 2011. We focus on more than 15 studies that are compared and evaluated in terms of the data they require, the methods used, their degree of automation, their metric performances, and their strengths and weaknesses.

**Keywords:** tsunami; artificial intelligence; remote sensing; neural networks; damage assessment





## 1. Introduction

The 2011 Tohoku-Oki Earthquake was a 9.1 undersea megathrust earthquake that occurred on Friday of 11th March 2011 at 2:46 p.m. local time (JST). It triggered the subsequent powerful Tohoku tsunami, with waves reaching up to 40 m, which wreaked havoc in the coastal towns of the Iwate, Miyagi and Fukushima prefectures, traveling as far as 5 km inland in the Sendai area [1]. This disaster caused more than 15,600 deaths, and 10 years later the Japanese government is still assessing the full extent of the damage caused to households and infrastructures [2]. On 10th December 2020, the estimate is that 121,992 buildings were totally collapsed, 282,920 half collapsed, and another 730,392 buildings were partially damaged in the Tohoku region.

While tsunamis are rare events compared with other natural disasters, they are particularly devastating. Furthermore, assessing the damage in the aftermath is a difficult task that must be done within a limited timeframe, as well as limited resources and information on site: the necessary logistics to conduct field surveys are complex and should be avoided, to prevent human exposure to hazardous areas. However, in the wake of such a disaster, accurate damage mapping is a race against the clock: the types of damage that need to be mapped the fastest are collapsed buildings and flooded areas, with the goal of quickly dispatching help to the right places and ultimately saving lives [3].

Unlike previous, similar disasters (such as the 2004 Sumatra-Andaman earthquake and tsunami), for the Tohoku tsunami, a fair amount of available high-resolution satellite images were available for the first time [4]. As such, it was also the first time that data scientists and geologists were able to use artificial intelligence algorithms to make an attempt at an automated, detailed, and high-quality assessment of the damage. Indeed, with the increasing availability of remote-sensing images and the rise of powerful artificial intelligence algorithms since 2011, the Tohoku tsunami has been a case study for many

researchers trying to propose automated fast-response algorithms and automatic damage-assessment methods, with the goal of providing key tools for disaster risk-reduction frameworks [5]. In particular, optical images and radar data from the Sendai bay area and from around the Fukushima nuclear plant have been used the most in these case studies; see Figure 1.

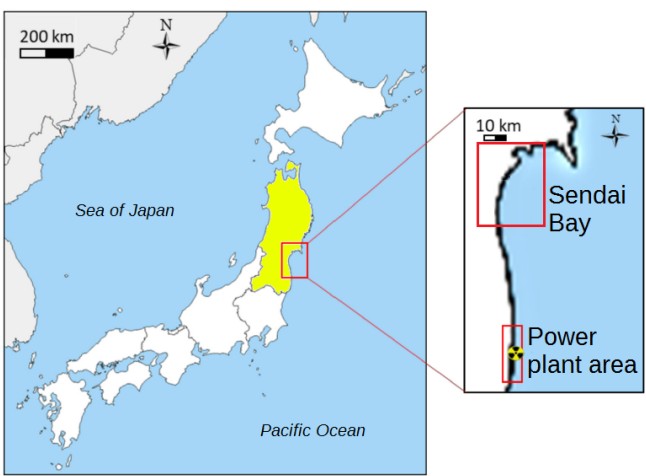

**Figure 1.** Zoom-in on the Tohoku region (in yellow), and the main study areas seen in the literature.

Within this context, this paper aims to provide a review of the different artificial intelligence (AI) algorithms and frameworks that have been developed for damage assessment based on case studies around the Tohoku Tsunami. In particular, we want to highlight how these methods have evolved through time, from simple index computation and manual mapping using thresholds, to powerful Deep Learning Algorithms that require high-precision images and data. Our review can be seen as complementary to the one from Koshimura et al. [6], in which the authors discuss different advances in remote-sensing technologies and their impact on the study of tsunamis in several aspects, such as the physics and acquisition of tsunami features, manual and automated damage interpretation with different modes of acquisition, and global response frameworks from remote-sensing images. In addition to a broader coverage of recent methods, our review focuses only on the artificial intelligence aspects, with a stronger focus on Deep Learning, and goes into the details of a dozen recent algorithms for automated damage assessment. Not only do we describe these methods in detail, we also analyze the strengths and weaknesses of the different AI techniques used: we assess their merits from a methodological perspective as well as from their quantitative and visual results, but we also discuss eventual weaknesses in the experimental protocols, such as the evaluation criteria used and eventual biases and limitations that may stem from it. We also address the degree of automation of the methods presented, the data they use based on the sources available at the time, and to what degree these methods could be re-used for a live tsunami event today. Finally, despite tremendous progress, we point out a gap between the current state-of-the-art in machine learning and the method used in the Tohoku tsunami studies. We highlight the relatively weak collaborations on the subject between the geoscience and remote-sensing communities, and the computer science and artificial intelligence ones. As a result, our review of the literature indicates that the former community under-exploits existing AI methods and still heavily relies on manual data curation and basic, on-the-shelf AI algorithms, while the latter community lacks the field knowledge to apply state-of-the-art AI algorithms in the ways that would be most useful for real applications. We encourage potential readers to consider reading both reviews if they are interested in obtaining a broader picture of the subject, and to focus on the review from Koshimura et al. [6] for a more application-focused perspective, or if they want details on the damage-assessment methods that are not covered in this work. Another, and certainly more modest, contribution is that, in this paper, we

try to unify the different notations and terms used for the different categories of damage, especially the terminology of building damage, which varies greatly between studies.

The remainder of this paper is organized as follows: In Section 2 we present the main source of data used by artificial intelligence in tsunami damage assessment. In Section 3, we review the different artificial intelligence methods that have been applied for automated damage assessment in images from the Tohoku tsunami: First, we introduce the different types of approach, the classes of interest and the evaluation criterion. Then, we detail a dozen of these AI methods, with a stronger focus on deep-learning ones, and analyze the results. Finally, this paper ends with a discussion and some conclusions on the progress made in the last 10 years and the current limitations that we face when using AI for damage assessment.

## 2. Remote Sensing Data: A Complex Source Material for Tsunami Damage Interpretation

When it comes to assessing the damage caused by the Tohoku tsunami using artificial intelligence methods, remote-sensing data were the source material of choice, as they proved to be faster and less risky to acquire than ground in situ data such as visual inspections, while also being easier to exploit on a large scale.

While they can involve either aircraft, drones or satellites, remote-sensing images are still most commonly acquired by artificial satellites with various sensors on board. These sensors can be divided into two categories: active and passive. Active instruments use their own source of energy to interact with an object. On the other hand, passive instruments use the energy emitted from natural sources, the sun in most cases.

Active sensors usually refer to the Synthetic Aperture Radar (SAR), which measures the roughness of surfaces. The main idea of this approach is to measure surface backscattering: the portion of the radar signal that is redirected back by the target.

Passive sensors, on the other hand, are mostly made of optical sensors that measure the amount of sun energy reflected by the target. For these optical sensors, we exploit different radiation wavelengths (or frequencies) from the electromagnetic spectrum (some of which match with visible colors). Figure 2 shows the electromagnetic spectrum with the corresponding wavelengths.

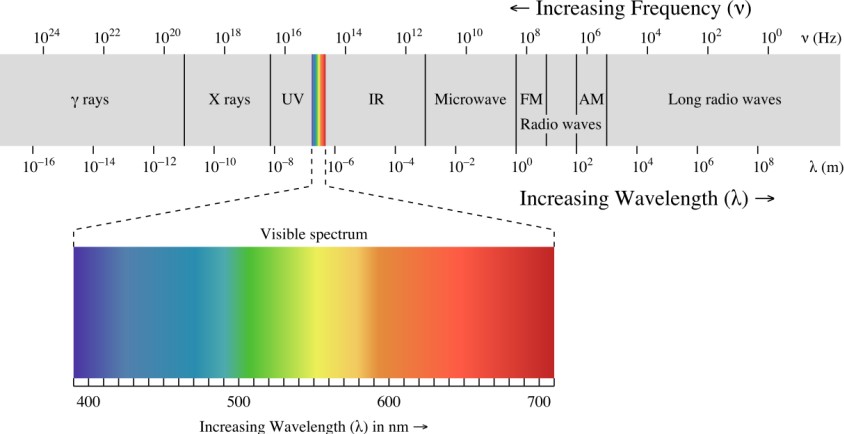

**Figure 2.** Electromagnetic spectrum.

Data from both active and passive sensors have been used, alone, together, and combined with aerial images and Light Detection And Ranging (LIDAR) data, to assess the damage caused by the Tohoku-Oki Tsunami.

In Table 1, we display the characteristics of the different missions and sensors, alongside the different studies in which images or data from these missions were used. The following abbreviations are used in this Table:

- Regular red, green and blue (RGB) sensors typically used by drones or aircraft to take pictures;
- Visible and near Infared (VNIR) sensors produce RGB images with an extra near infrared channel;
- Panchromatic (PAN) images include the four channels from VNIR images and extra channels that combine different colors. In the case of the Worldview missions, the four extra channels are: red edge (between red and near-infrared), coastal (between blue and ultra-violet), yellow (between green and red), and near-IR2 (a second near-infrared channel with a higher wavelength);
- Polarimetric L-band Synthetic Aperture Radar (PoLSAR).
- Thermal Infrared (TIR).

In the following subsections, we will detail the advantages and disadvantages of the different modalities of acquisition, depending on the type of damage being detected.

**Table 1.** List of missions whose data were used for case studies on the 2011 great tsunami. The first part of the table shows satellite missions, and the second part shows other airborne options.

| Mission | Active Years | Spatial Resolution | Sensors | Studies |
|---|---|---|---|---|
| Infoterra GmbH/TerraSAR-X | 2010–now | 0.5–40 m | SAR | [7–15] |
| DigitalGlobe/Worldview-2 | 2009–now | 2 m | PAN | [16,17] |
| ASI COSMO-SkyMed | 2007–now | 1–30 m | SAR | [18,19] |
| JAXA ALOS/AVNIR-2 | 2006–2011 | 10 m | VNIR | [20,21] |
| JAXA ALOS/PALSAR | 2006–2011 | 7–89 m | PoLSAR | [9,13,22–25] |
| ESA/ENVISAT | 2002–2012 | 260–300 m | VNIR | [26] |
|  |  | 28–30 m | ASAR |  |
| NASA/ASTER | 1999–now | 15 m | VNIR | [26,27] |
|  |  | 90 m | TIR | [26] |
| NICT/Pi-SAR2 (airborne) | 2011–now | 0.3–1 m | SAR | [22] |
| Other aircrafts & drones | - | - | LiDAR | [28,29] |
|  | - | - | RGB | [7,20,21,28–31] |

### 2.1. Optical Images as Source Data

Optical images are the ones that we are most familiar with, and they are the closest to what is used in other domains of artificial intelligence. With these images, it is relatively easy to adapt existing algorithms, such as traditional segmentation methods [32], deep learning and convolutional neural networks [33,34], or couplings between segmentation methods and classification algorithms. Furthermore, with their extra channels, these images offer the possibility of computing several indexes such as the normalized difference vegetation index (NDVI), the normalized difference soil index (NDSI), or the normalized difference water index (NDWI) [35]. Computing these indexes on images from before and after a tsunami can be very useful to detect flooded areas, washed-up buildings or destroyed vegetation.

For example, in [20,21], the authors use the NDWI to assess which areas were inundated by the 2011 tsunami based on images from the ALOS program [36] of the Japanese Aerospace Exploration Agency (JAXA).

Nevertheless, optical images also have their limits and weaknesses. The first, obvious one is the resolution of the images, which, as can be seen from Table 1, varies greatly from one mission to another and can sometimes be insufficient to detect small buildings (See Figure 3). The second issue is the differences in lighting that may occur with images taken under different atmospheric conditions, at different times of the day, or in different seasons. This problem had to be taken into consideration, especially for approaches that use pairs of images from before and after the 2011 tsunami, as this proves to be problematic for things

as simple as index comparisons, but also for more complex comparisons of textures and analysis using artificial intelligence. Furthermore, optical images also suffer from cloud occlusion and shadows [37]: the presence of clouds and their shadows makes the analysis of optical images from difficult to impossible, and sometimes forces researchers to pick another image farther away in time, thus increasing the risk of strong seasonal differences. It is also worth mentioning that, in 2011, images were taken less frequently. As a result, flaws or issues with an image could mean that the closest available one was from several months before or after.

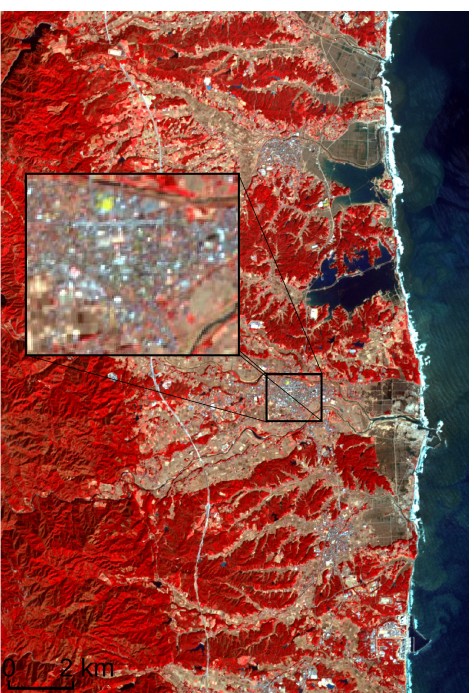

**Figure 3.** ASTER Image with false colors of the Fukushima Prefecture taken on 19 March 2011. The zoom on this image shows that the resolution might not be high enough to individually detect the status of small buildings.

These problems are described by the authors of [27], who used a before and after approach based on available ASTER images of the Fukushima Prefecture, and had trouble finding an image from before the tsunami that was clear of clouds and without too many seasonal effects compared with the aftermath image. As one can see from Figure 4, Sub-Figure a shows an example of extreme seasonality effects, while Sub-Figure b, which is closer to the disaster, suffers from cloud occlusion.

Finally, in [16], the authors mention a lesser-known issue in optical images from the Tohoku tsunami: their work addresses the evolution of the shoreline caused by the tsunami using Worldview-2 images, and points out the issues caused by foam and waves when making a proper coastal delineation. The same foam pixels in the same area between Ukedo harbor and the Fukushima nuclear reactor, see Figure 4c, were mentioned by the authors of [27] as causing issues with the unsupervised methods they applied to ASTER images. Indeed, unsupervised algorithms seemed to not know which clusters the foam should be assigned to, and tended to pair it with destroyed buildings.

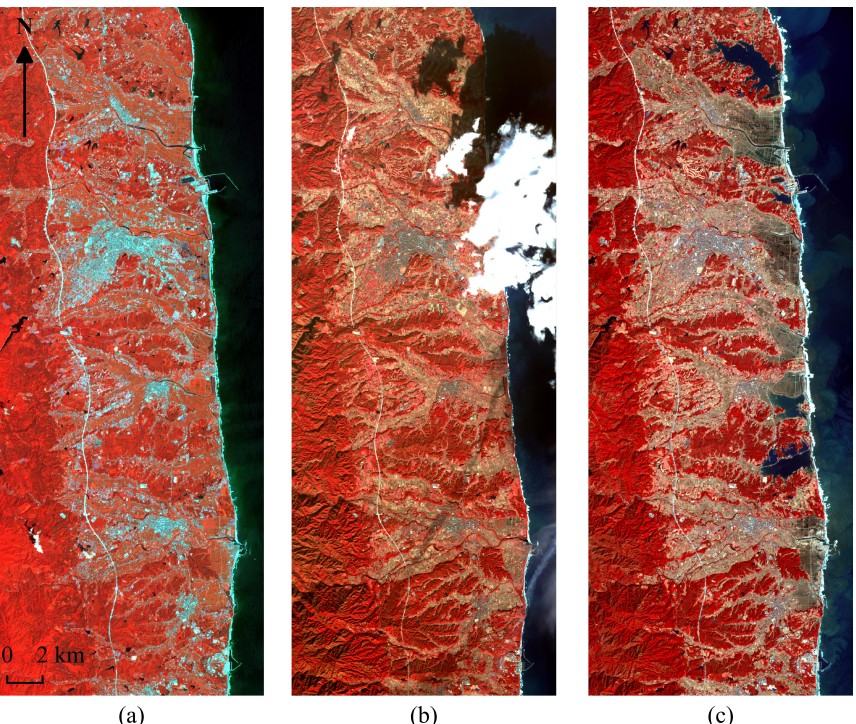

**Figure 4.** ASTER Images in false colors of the Fukushima prefecture showing an example of seasonality effects in the vegetation and cloud occlusion issues [27]: (**a**) was taken on 10 July 2010, (**b**) on 29 November 2010 and (**c**) on 19 March 2011.

### 2.2. Synthetic Aperture Radar Images as Source Data

One solution to the problem of cloud occlusion and lighting issues faced by optical images is to use the SAR data acquired by satellites. Indeed, radar sensors use longer wavelengths at the centimeter-to-meter scale, which gives them special properties, such as the ability to see through clouds and to operate in both day and night conditions. To map surfaces, radars rely on information such as the strength of the signal reflectance, signal backscattering, and phenomena such as double bounce, all of which are particularly useful to analyze the textures of the different surfaces observed by the radar. In particular, just as for optical images, certain land cover surfaces, such as water, forests areas and urban areas, have known backscattering and reflectance histograms. For example, in [6,38], the authors used the reflectance and signal backscattering histograms to detect water bodies, which may indicate the ocean, but also flooded areas. Numerous AI approaches have also been used to detect damages at the building-unit scale, most of them relying on TerraSAR-X images [7,8,10].

While they have the advantage of not being affected by clouds or light conditions, SAR images have also weaknesses. Among them, fully polarimetric SAR data are not always available, and, as can be seen from Table 1 their resolution in 2011 tended to be lower than the resolution from optical images [23], and this is especially true for the L-Band. Furthermore, in addition to their often lower resolution, these data suffer from *speckle noise*, which an issue for approaches that rely on comparisons of two images from before and after the disaster.

Taken together, the two previously mentioned weaknesses imply the importance of pre-processing the images to reduce the speckle noise [39–41], and potential difficulties in properly detecting, segmenting or delineating small elements. This has been proven to be a problem when detecting damage at the building level for lower-resolution images [42], but it does not prevent the broader categorization of the different levels of damage that buildings can suffer from a tsunami over large areas [13], namely being slightly damaged, collapsed or washed away.

To conclude on SAR data, it is worth mentioning that not all of them are acquired from satellites. Some come from aircraft and drones, such as the PI-SAR2 data [43] that were used on a study of damage assessment [22], and have a higher resolution. However, this program was launched in early 2011 and few data were available at the time. Furthermore we will see in the next subsection that the use of aircrafts (especially manned ones) raises other issues.

### 2.3. Aerial Images and LiDAR

Given the cost of satellite missions and the issues that we mentioned before, such as gaps between the images and low-resolution problems, post-tsunami damage assessment frequently uses aerial images and LiDAR data as a complementary source of information.

Aerial color images have many of the advantages of optical images and can be used to relatively quickly acquire high-resolution and high-quality RGB images of any area. The main weakness of this approach is that, while the resolution might be better, the area covered by these images is also a lot smaller compared with remote-sensing data. Furthermore, aerial images are subject to similar issues to optical images from satellites, such as cloud occlusion, and lighting issues. One final issue that may be relevant in the case of the Tohoku tsunami is the difficulty of sending manned aircraft over a given area when (1) the nearby infrastructures such as airports are destroyed, or (2) there is a risk of radiation leak on-site.

LiDAR data are another option to acquire data with aircrafts or drones, but the technology was not really used for the purpose of damage assessment back in 2011. Still, some attempts exist to monitor the tsunami-affected areas, focusing on the accumulation of debris generated by the tsunami inundation using both LiDAR data and aerial images [28,29].

Overall, while they are not great for assessing damage to large areas, LiDAR data and aerial images have still been used as complementary data for SAR and optical images acquired from satellites. In particular, the detection of small elements, such as crumbled buildings and floating debris, is one application for which they can be useful.

## 3. Machine Learning and Deep Learning Applied to Images of the 2011 Tohoku Tsunami

In this section, we detail the different artificial intelligence algorithms that have been applied to images of areas that were struck by the Tohoku tsunami. We will highlight how the algorithms have evolved through time following the evolution of AI techniques.

Before we start, we believe that it is important to define what qualifies as artificial intelligence and what does not. To write this review, we have focused on a few more than 15 case studies, listed in Table 2. What all these studies have in common is the goal of identifying different types of damage based on the different data sources listed in Table 1: flooded areas, damaged buildings, washed-up or destroyed buildings, debris detection, and shoreline modifications. What differentiates them (beyond the targets being detected) is their level of automation, and the use of regular *old school* artificial intelligence versus neural-network-based deep-learning approaches. In this review, we stick to a definition of artificial intelligence as being something that requires learning abilities.

Based on these criteria, we have discarded a number of studies where damage assessment after the Tohoku tsunami was done using methods that we consider not to have a proper learning process and, therefore, to not be artificial intelligence [7,13,18–22,28–30]. These methods rely mostly on image-processing methods such as: index computation, thresholds, manual classification from visual cues or in situ data, image comparisons, and various basic segmentation algorithms not followed by an automated classification process. Please note that this qualification as "simple image processing" methods does not imply anything about the efficiency of these methods, with some of them performing just as well as AI methods and sometimes also being faster. In fact, many of them were absolutely necessary to pave the way for the AI methods that came after and to build the ground-truths used by these AI methods. If you are interested, several of them are detailed in the review by Koshimura et al. [6].

**Table 2.** The different studies and methodologies used on the Tohoku tsunami images, sorted by category. The *Image Processing* line is not considered AI in this work.

|  | Before & After Approaches | Analysis on the Aftermath Image(s) |
| --- | --- | --- |
| AI based on supervised learning | [8–10,15,16] | [12,24,25] |
| AI based on unsupervised learning | [14,23,26] | |
| Supervised Deep Learning | [17,31,44] | [11,31] |
| Unsupervised Deep Learning | [27] | |

As one can see, we have sorted the remaining AI methods into different sub-categories in Table 2, where the complexity of the methods increases from the top to the bottom and from left to right:

- Supervised and unsupervised methods;
- Methods that need images from *before and after* the disaster (including change-detection approaches), versus methods that only use images from after the tsunami;
- Regular learning approaches and deep-learning approaches

In Artificial Intelligence and Machine Learning, we distinguish supervised and unsupervised learning methods, and this distinction is often a good clue as to their degree of automation: in supervised learning, the algorithms have to be trained using user-labeled data that are fed to the AI algorithm so that it can later propose a classification for new data without labels. In unsupervised learning, the algorithm will explore the data on its own to find patterns or groups of similar elements that it finds remarkable. From an application perspective, supervised learning tends to achieve better performances. However, it requires labeled data to be available. As it turns out, such labeled data are rarely available right after a tsunami, when the mapping of damage without such data is urgent [14]. They have to be either produced quickly over a small area, or transferred from a previous disaster, both of which often proves difficult if not impractical. On the other hand, unsupervised learning tends to give lower performances but can be used on raw data without the need to provide any annotated data to train the algorithm. This is why unsupervised learning is considered to be more automated. It is worth mentioning that the automation of unsupervised learning has limits and that human intervention is still required to link the clusters found to actual classes of interest, which carries the risk that these clusters are not pure and encompass more than one class [27]. Furthermore, some machine-learning methods can be considered to be inbetween (semi-supervised) if they can benefit from a mix of labeled and unlabeled data. Obviously, in the case of quick damage assessment after a disaster, the fewer annotated data are needed, the better.

The second important split is between methods that assess damage based on images pre- and post-disaster, and the ones that only use images from after the tsunami. There are three reasons to use pairs of images from before and after the disaster:

- The first one is the intuitive idea of change detection between the two images by observing their differences. This well-known change-detection process is widely used in remote sensing [45]. In the case of tsunamis, it can be used as a damage assessment method on its own by simply computing changes in various indexes and textures. However, this can also be a pre-processing step to exclude areas of the images without interesting changes, so that the damage-assessment mapping algorithm can focus on the most critical area;
- The second reason consists of concatenating the two images and to using the newly created multi-channel image as a single input for an artificial intelligence algorithm. This methods enables the production of higher-quality features and textures;
- Finally, the pre-disaster images can be favored to map the pre-existing buildings (which may not be in the after images), and a heavier weight can be given to the post-disaster image to categorize the damage.

All three uses can be badly affected by the speckle noise of SAR data, as well as trivial and seasonal changes in optical images. As such, the pre-processing and the choice of the change-detection algorithm in the former, and the damage-assessment method in the latter, are key to mitigating these issues.

On the other hand, since pre-disaster images are not always available, a second approach takes the stand that the post-disaster images should be self-sufficient and that the damage can be assessed by segmenting and classifying the different elements. In practice, this is often proven to be only partially true, at least for areas with water: the pre-disaster images or a reference dataset are needed to differentiate flooded areas from rivers, lakes and the sea [7].

Finally, a distinction can be made between regular machine-learning methods, and more recent methods that rely on deep neural networks and, in particular, convolutional neural networks (CNN) [33,34]. The former are simpler AI methods that are usually more than 30 years old, but have proven fast and effective for many problems and can be run with a low number of training data. On the other hand, more recent deep-learning methods are usually slower, and require more data (labeled in most cases), but give results that usually outperform the ones from regular AI methods by far. One common family of deep learning methods that we will mention in this review is convolutional neural networks. They are made of stacked layers of convolution filters, linked by activation functions and pooling layers, which are used to extract textures and high-quality features. They usually also contain some linear layers at the end for classification purposes [46].

Please note that all the previously presented categories are mutually compatible. For instance, convolutional neural networks have been shown to be extremely good for comparisons between from two images [47].

### 3.1. Classes of Interest and Metrics

Before we begin with the different AI methods used on studies for the 2011 Tohoku tsunami, in Figure 5, we have summarized the main classes of damage (or no damage) that are usually considered by the different authors applying artificial intelligence for damage assessment. In this figure, we make an attempt to unify and create a hierarchy between the different notations—especially for building damage—that can be found in the different studies, as not all authors go into the same level of detail. Furthermore, the Figure also proposes a color code from green to brown, which gives an idea of the seriousness of each class of damage.

The correct or incorrect detection of these classes is commonly evaluated using different metrics, such as the accuracy, the F1 score or the Kappa index. The accuracy and F1 score are dice binary indexes that rely on the notion of true positive (TP), false positive (FP), true negative (TN) and false negative (FN). They can be computed for a given individual class, or an average can be computed for all the considered classes

$$Accuracy = \frac{TP + TN}{TP + FP + TN + FN} \tag{1}$$

$$F1 = \frac{TP}{TP + 0.5 \times (FP + FN)} = 2 \cdot \frac{prec \times recall}{prec + recall} \tag{2}$$

The accuracy (or overall accuracy, if considering several classes) tends to be the most commonly used metric, as it is the most intuitive one. However, it is weak regarding imbalanced classes and does not penalize strong false negative misclassifications. The F1 score, on the other hand, handles class imbalance well and give a better measure of incorrectly classified cases than the accuracy metric. In artificial intelligence, it tends to be the most-used metric, as it gives the best evaluation of real-life problems. We can also mention that the F1 score is the harmonic mean between the precision and the recall, two other dice metrics that emphasize the percentage of positive identification that was correct and the percentage of correctly identified positive cases, respectively. Sadly, these two

latter indexes, although very useful to assess the weaknesses of a classifier, are rarely used in studies of damage assessment.

$$Precision = \frac{TP}{TP + FP} \quad , \quad Recall = \frac{TP}{TP + FN} \tag{3}$$

The Kappa index, on the other hand, was originally designed as a statistical tool measuring the agreement between two models or observers. In classification, it is often used to measure the agreement between the result of an algorithm and the ground-truth. Unlike the dice indexes, the Kappa index takes the possibility that two models may agree by chance into consideration. As such, it is considered more reliable when dealing with tasks in which class imbalance is an issue. It is computed as shown in Equation (4), where $P_o$ is the relative observed agreement (equal to the accuracy), and $P_e$ is the hypothetical probability of chance agreement, using the observed data to calculate the probabilities of each observer randomly picking each category.

$$\kappa = \frac{P_o - P_e}{1 - P_e} \tag{4}$$

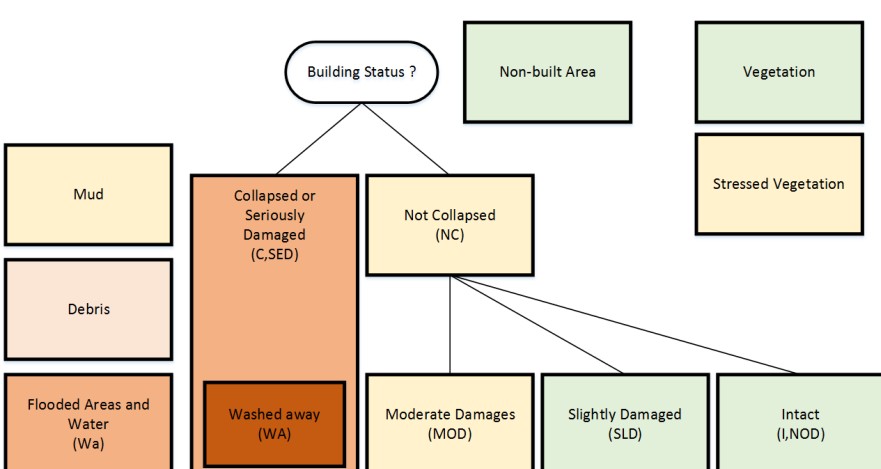

**Figure 5.** The main classes of damage that can be found in the artificial intelligence (AI) literature applied to the 2011 Tohoku tsunami. Note that, for building damage categories, there seems to be a hierarchy, and the "washed away" category can even be considered as a specific subset of the "collapsed building" class.

### 3.2. Approaches Based on Regular Machine Learning

Since several of the regular machine learning approaches are already presented in detail in the review by Koshimura et al. [6], in this section, we mostly detail the regular AI approaches that are not presented in their review. However, not all methods presented in Koshimura et al. were discarded, as we feel that some of them are important for methodological comparison purposes.

### 3.2.1. AI Based on Supervised Learning

We begin with regular AI approaches using supervised learning algorithms. In [10], Endo et al. proposed a multi-class damage-classification method for buildings based on TerraSAR-X images from before and after the tsunami. Their focus area is the coastal area of Sendai. After pre-processing the images from before and after the tsunami, they extract features for damage classification, which are then fed to Support Vector Machines (SVM) [48]. The authors use both a map of the building footprints to detect lateral shift, and a ground-truth for the damage assessment training. After trying to recognize seven degrees of damages that were too difficult to properly distinguish with their SVM model, the authors settled for three classes: slight damage, moderate damage and washed away.

By doing so, they achieved a mean F1 score of 71.1%. The results are shown in Figure 6. As one can see, the classification seemed to be more difficult in the Sendai harbour area than in Sendai Wakabayashi. This is most likely due to the sturdier nature of the industrial buildings in the harbor area, which makes it more difficult to detect damages from the sky, thus leading to an underestimation of their deterioration status.

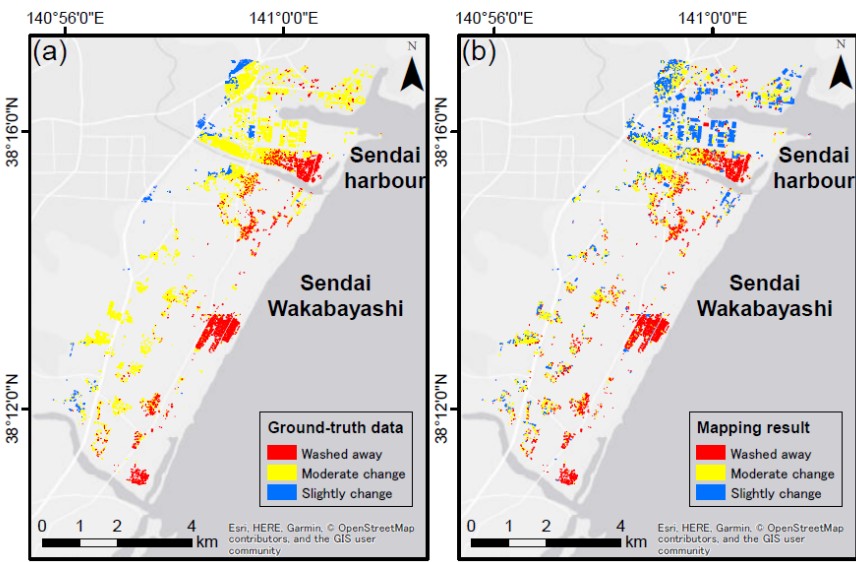

**Figure 6.** Visual results for Endo et al. SVM approach to map building damage. (**a**) Ground-truth; (**b**) mapping results. The image is from Endo et al. [10] and re-used with the corresponding author's permission.

In [9], Wieland et al. propose to compare the performances of SVMs for tsunami damage classification against the use of simple thresholding methods based on TerraSAR-X and ALOS PALSAR images from before and after the disaster. The goal of their study was to show that a simple Machine Learning algorithm like SVM would work better than the, widely used at the time, change image thresholding. Their proposed architecture is very simple and relies on pre-segmentation using existing blue prints before feeding the data to the SVM algorithm. They also tested different data sampling techniques in order to assess the sampling influence on the final results. They concluded that balanced random sampling gave the best results, with a F1 score of 0.85 for change detection. This study is interesting, but sadly it does not go beyond the detection of change or no-change areas, thus not providing detailed categories of damage. Nevertheless, it is the first study that showed the superiority of Machine Learning compared to manual thresholding methods.

Another SVM model was proposed by [12] on the same dataset. It relies on different input features built from the gray-level co-occurrence matrix (GLCM) in order to obtain better textures. The novelty in this study is that the GLCM is constructed in three dimensions. The authors have proven that this 3DGLCM tended to be a near-diagonal matrix in areas without damage, whereas the non-zero components were far from the diagonal in damaged areas, thus leading to better performance in identifying collapsed buildings with overall F1 scores between 80% and 91%.

In [24,25], Ji et al. also propose another SVM-based method based on PolSAR images from after the tsunami. Unlike what the authors claim in their abstract, the proposed method is clearly not unsupervised, since it heavily relies on a first land-cover classification achieved with the SVM algorithm, as can be seen in Figure 7. The main problem tackled by this paper is that the Miyagi Prefecture after the disaster comprised a complex situation in which damaged buildings with various—and potentially very different—orientation angles existed, and were wrongly labeled as undamaged by conventional methods. To solve this problem, they propose classifying the whole urban area into parallel buildings

and oriented buildings by setting a threshold based on the circular correlation coefficient, so that damaged buildings can be identified from both building categories.

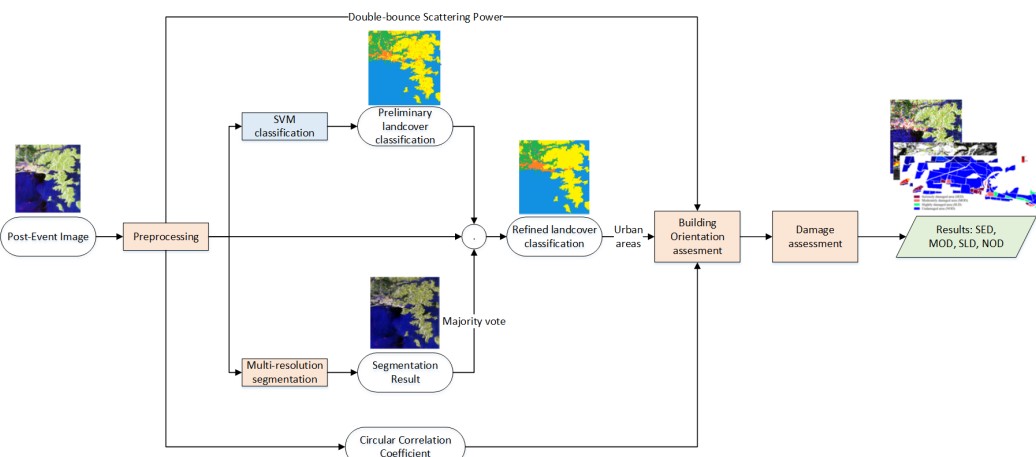

**Figure 7.** Flowchart of the proposed architecture proposed by Ji et al. The miniature images are from [24].

Their full methodology is the following (cf. Figure 7): First, the raw post-disaster PolSAR image is pre-processed by applying multi-look processing and a polarimetric Lee refined speckle filter [49]. Then, a preliminary classification into four classes (water, mountain, farmland, and urban) is done using SVM fed with POA-compensated polarimetric features. Because the damaged buildings and oriented buildings shared similar polarimetric characteristics to the foreshortened mountainous areas, as described by eigenvalue–eigenvector-based polarimetric decomposition, the authors used a threshold criterion and segmented-region-based majority voting to compute an improved region-based classification result. The threshold criterion is based on the polarimetric coherence magnitude and the sum of eigenvalues from Cloude–Pottier decomposition [50]. Then, the segmentation is carried out using a multi-resolution segmentation technique. Finally, once this classification step is over, the damage assessment itself could be considered unsupervised since it does not rely on training data, and is done based on manually set thresholds and the use of the circular correlation coefficient. However, in the result, the AI part is focused on predicting the land cover, and not at mapping or assessing the damage. The authors considered four levels of building damage: serious damage, moderate damage, slight damage, and no damage (matching with the intact class from Figure 5).

The results of their methods when applied to Ichinomaki city are shown in Figure 8. They achieved a 92.28% accuracy of correctly classified blocks, compared with 88.81% for the SVM algorithm alone. Ji et al.'s proposition clearly outperforms the SVM algorithm alone in the proper detection of seriously and moderately damaged buildings, but both are equally bad at differentiating between slightly damaged buildings, and buildings with no damage. This is most likely due to the difference being impossible to detect from the sky with SAR images. It is worth mentioning that the block approach used in the experimental results, and the use of a manual threshold between Figure 8a,b, might have artificially pulled-up the accuracy metrics, thus providing Ji et al. propositions with an unfair advantage compared with other studies presented in this review.

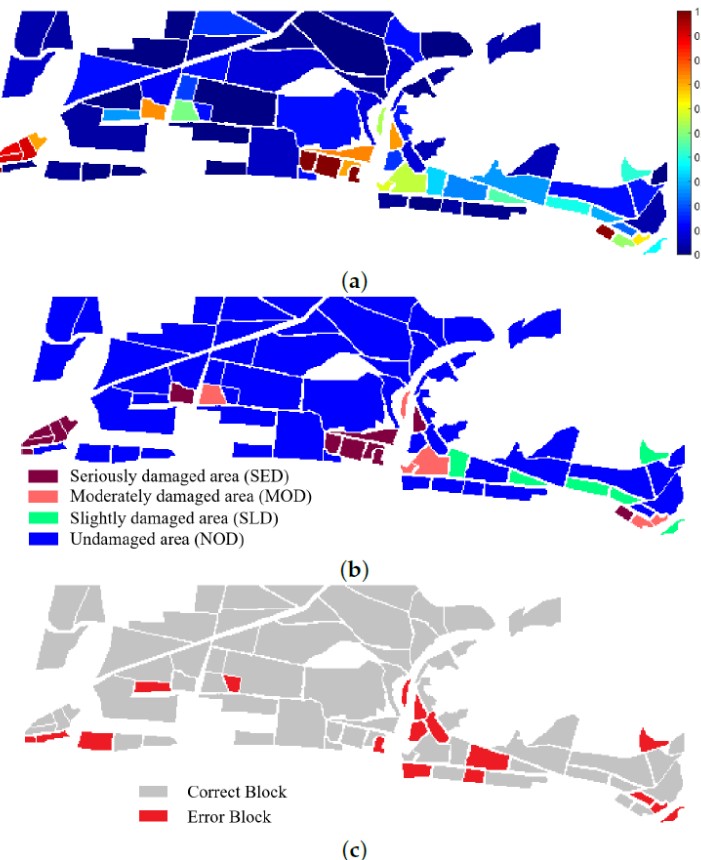

**Figure 8.** Results of the method proposed by Ji et al. on Ichinomaki city (**a**) Damage at the block level using their proposed method (**b**) Bloc classification into 4 classes using their method (**c**) Block-based error map; The image is modified from [24] with the authors' permission.

The study proposed in [16] is different to the others presented in this review in the sense that it does not map damage caused to the buildings or inundated areas, but it assesses damage caused by the Tohoku tsunami to the coastline itself in the form of erosion or accretion. To do so, the authors used an SVM applied to Worldview-2 images of the coastline in the Fukushima Prefecture and Sendai bay area taken before and after the tsunami. In this work, the authors aim to remove the noise caused by foam—generated by both waves and debris—so that the coastline can be detected properly, thus enabling the observation of any change that might have been caused by the tsunami. The authors tried several models, which achieved less than 1% false-positive and less than 2% false-negative detection in properly detecting the land, foam and water classes. Based on these results, they estimate the coastline erosion and accretion in both areas with an estimated surface relative error that outperformed other machine-learning approaches, including neural networks.

### 3.2.2. AI Based on Semi-Supervised Learning

In [14], the authors propose an approach called the *Imagery, Hazard, and Fragility function method* (IHF), where the idea is to use the experience gathered from various previous disasters such as known correlation between the disaster intensity and the damage, so that this information can be used to tune Machine Learning algorithms instead of a long and tedious manual map annotation. Their method is based on a modified logistic regression model—a normally supervised algorithm used for binary classification—that the authors have modified so that the parameters that are normally trained using labeled data can be approximated without them (as they are often unknown) using an adequate fragility function [51,52].

In their paper, the authors use a fragility function that can be computed by knowing the depth of the inundation that can be assessed from ground measures or by pre-processing the SAR images [13]. They used the curves proposed by Koshimura et al. [53], as well as Suppasri et al. in [54], to produce and refine an adequate fragility function; see Figure 9. In other words, the supervision and the training data are replaced by an intensity parameter (e.g., the inundation depth) and probabilistic information computed from expert knowledge through the fragility function.

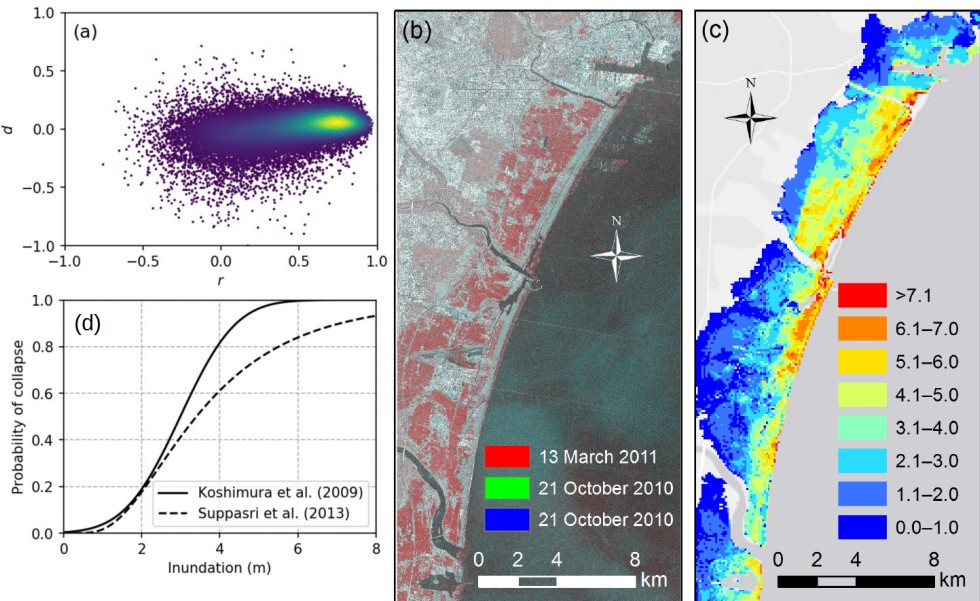

**Figure 9.** (**a**) Scatter plot of the data using the average difference in backscattering between the disaster intensity *d* and the correlation coefficient *r*. (**b**) RGB composite image made of TerraSAR-X data acquired on 13 March 2011, and 21 October 2010. (**c**) Known innundation map used for the disaster intensity. (**d**) Empirical fragility functions of buildings proposed by Koshimura et al. [53] and Suppasri et al. in [54]; The figure is modified from [14] with the corresponding author's permission.

They apply their method to TerraSAR-X images of the Sendai bay area taken on October 12th 2010 and March 13th 2011. They use input features of the form $x_i = \{1, d, r\}$ where *d* is the average difference in backscattering between the two images, and *r* is the correlation coefficient. They try to predict two classes of buildings—collapsed and not collapsed—and manage to do so with an accuracy of 87.5%. Furthermore, based on scatterplots such as the one shown in Figure 9d), the authors also attempted to find regression lines with the goal of distinguishing different degrees of damage caused to the buildings based on data from the Japanese Ministry of Land, Infrastructure, Transport, and Tourism [55].

The approach used in this paper is interesting in the sense that the basic model is simple, yet effective, and can easily be adapted to tsunamis in other areas using the different fragility functions that are available for several countries, such as Chile [56], Thailand [57], Sri Lanka [58], and the Samoa islands [52,59]. However, from an AI perspective, this type of methodolody, relying on expert functions, should be considered semi-supervised learning rather than fully unsupervised. Furthermore, given the small number of available disasters for which these functions have been designed, another pitfall could be the controversy regarding whether or not these empirically built functions are fully transferable.

In [15], the authors use the same combination of a fragility function with a machine learning method, this time with Support Vector Machines. They also introduce the demand parameter using various thresholding techniques and field information from in situ sensors, which allow to better define high-priority areas.

### 3.2.3. AI Based on Unsupervised Learning

We now move on to unsupervised learning methods. In [26], the authors use the K-Means algorithm [60], which is probably the most well-known and simple unsupervised AI algorithm. Its principle is very simple: given the *K* prototypes defined in the same space as that data, which are randomly initialized, the algorithm alternatively links each datum to its closest prototype, and then updates the prototypes as the mean of the datum linked to it, and the process is repeated until convergence.

In their application, Chini et al. [26] used the K-Means algorithm on pre-processed features extracted from ASTER VNIR and TIR images taken before and after the tsunami. Since the K-Means algorithm requires a manual setting of the number of clusters, and tends—like many clustering algorithms—to fail to exactly match the classes of interest, the authors set up the algorithm with $K = 30$ clusters and appear to have manually merged them afterwards: the 30 clusters were grouped into 1 *no change class* and 6 *change classes*, namely: flooded areas, debris, damaged buildings, mud, stressed and non-stressed vegetation. While this method has the advantage that the cluster generation is fully automated and relies on a fast algorithm (K-Means is linear in $O(N)$ for a dataset of size $N$.), it has the following major issues: the 30 resulting clusters apparently need to be manually mapped to the classes of interest, with no guarantee that some of them will not be a mix of several classes. For such a scenario, where clusters have to be merged, hierarchical clustering [61] should be considered a better option, despite its higher time complexity. Finally, another problem with this study is that it does not provide any quality indication of the results in terms of accuracy compared with the ground-truth.

In [23], Park et al. use a similar change-detection technique based on the Expectation Maximization algorithm [62] coupled with Markov Random Fields [63] to reduce the speckle noise. The Expectation Maximization algorithm for gaussian mixtures is another unsupervised technique, similar to the K-Means algorithm, but with a variance covariance matrix that allows them to reach ellipsoid clusters instead of the spherical-only clusters of the K-Means algorithm. They used three groups of features acquired from SAR images acquired on 21 November 2010 and 8 April 2011: the first group was composed of the difference in the characteristics of the two images (surface, double-bounce and scattering angle) acquired through Cloude–Pottier decomposition [50]. The second group was acquired using a Polymetric SAR image decomposition method based on eigenvector decomposition [64]. The third group includes the polarimetric orientation angle, and the co-polarization coherence on the linear polarization basis and the circular polarization basis. They considered three classes, two different change classes for damaged areas and a no-change class. They achieve high-accuracy performances of around 90%.

### 3.3. Approaches Based on Deep Learning

In the previous subsection, we have presented some applications of very basic AI algorithms to the automatic damage assessment of the 2011 Tohoku tsunami based on remote-sensing images. These algorithms included SVM, as well as simple decision trees for the supervised methods, and K-Means, Expectation-Maximization, and a supervised logistic regression based on expert knowledge for the unsupervised and more automated methods.

In this subsection, we take a look at more advanced AI techniques, based on deep-learning algorithms that have been used for the same damage assessment problem.

### 3.3.1. Supervised Deep Learning

In [11], Bai et al. propose a deep-learning-based approach for damage recognition using SqueezeNet [65] and a wide residual network [66] applied to post-disaster TerraSAR-X images. SqueezeNet and wide residual networks are both light models with high performances, built to reduce the complexity of early CNNs [33] and residual networks [67], respectively. In their architecture, the authors start with a tile-splitting process, which is necessary to feed the image to their neural network. The data are also transformed from digital numbers to sigma nought and are pre-processed for speckle noise using Lee

filters [49]. The SqueezeNet is then used to separate tiles with buildings from tiles without buildings. Then, the authors apply their wide residual network to classify the damage inside built tiles into washed-away buildings, collapsed buildings and slightly damaged buildings. Both networks are trained using the manually annotated data available from the Japanese Ministry of Land Infrastructure, Transport and Tourism [68]. The full architecture, as well as the results, are shown in Figure 10.

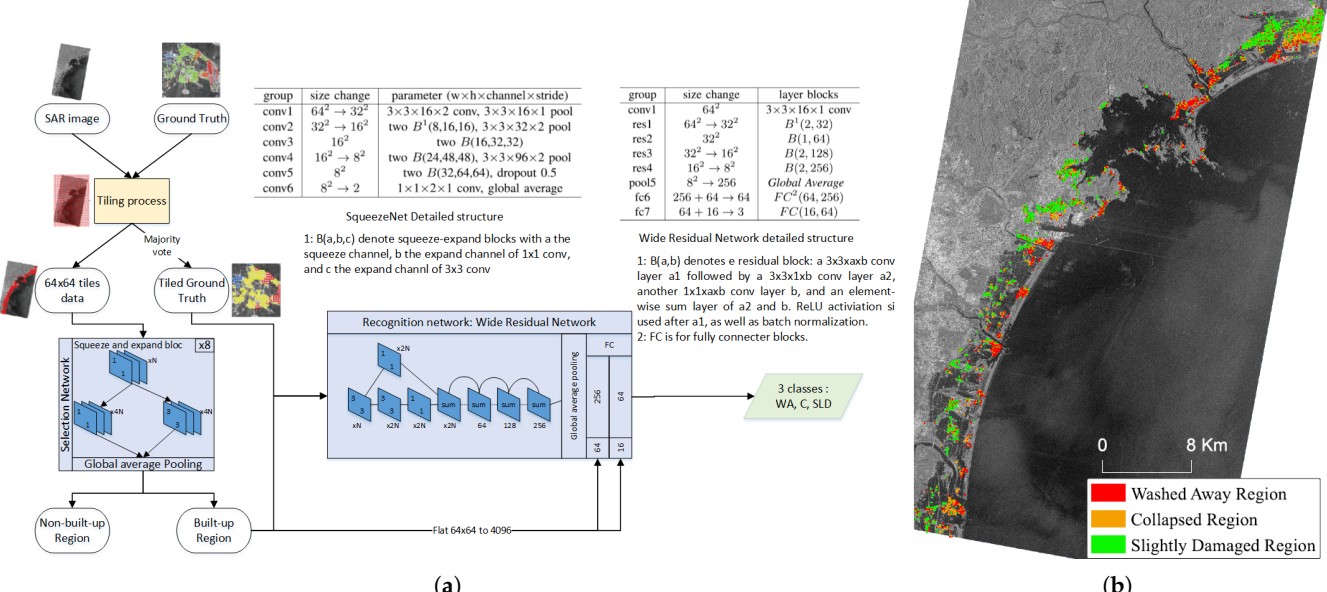

**Figure 10.** Application of the proposed modified Wide Residual Network by Bai et al. [11] to TerraSAR-X images. (**a**) Proposed architecture by Bai et al. relying on SqueezeNet and a Wide Residual Network. This Figure is inspired by the original paper [11]. (**b**) Visual results with 3 classes in the Sendai area; The image is adapted from [11] with the author's permission.

This paper is particularly interesting because the authors made a real effort to adapt an existing method to the problem of damage assessment, and proposed the comparison of different deep-learning methods, such as AlexNet [69] (a CNN that is heavy in parameters compared with SqueezeNet [70]) and an original residual network, in addition to several versions of their modified networks with different parameter configurations. With the best configuration, their network achieves around 75% accuracy.

In [17], the authors propose another supervised damage-mapping method, this time inspired by the well known U-Net algorithm [71]. Their algorithm is powered using the Microsoft Computational Network Toolkit [72]. U-Nets are a family of deep convolutional neural networks, originally proposed for medical images, which have the ability to segment images, and then sort the segments into classes using a supervised learning process. In addition to having additional batch normalization [73], the authors have proposed other modifications to the original U-Net architecture, including the replacement of max-pooling layers by convolutions of stride 2, and a complete reshaping of the upward branch of the U-Net. The result is unlike other U-Nets, since the number of features does not seem to increase with depth, and the network is no longer symmetrical. The full architecture is presented in Figure 11a.

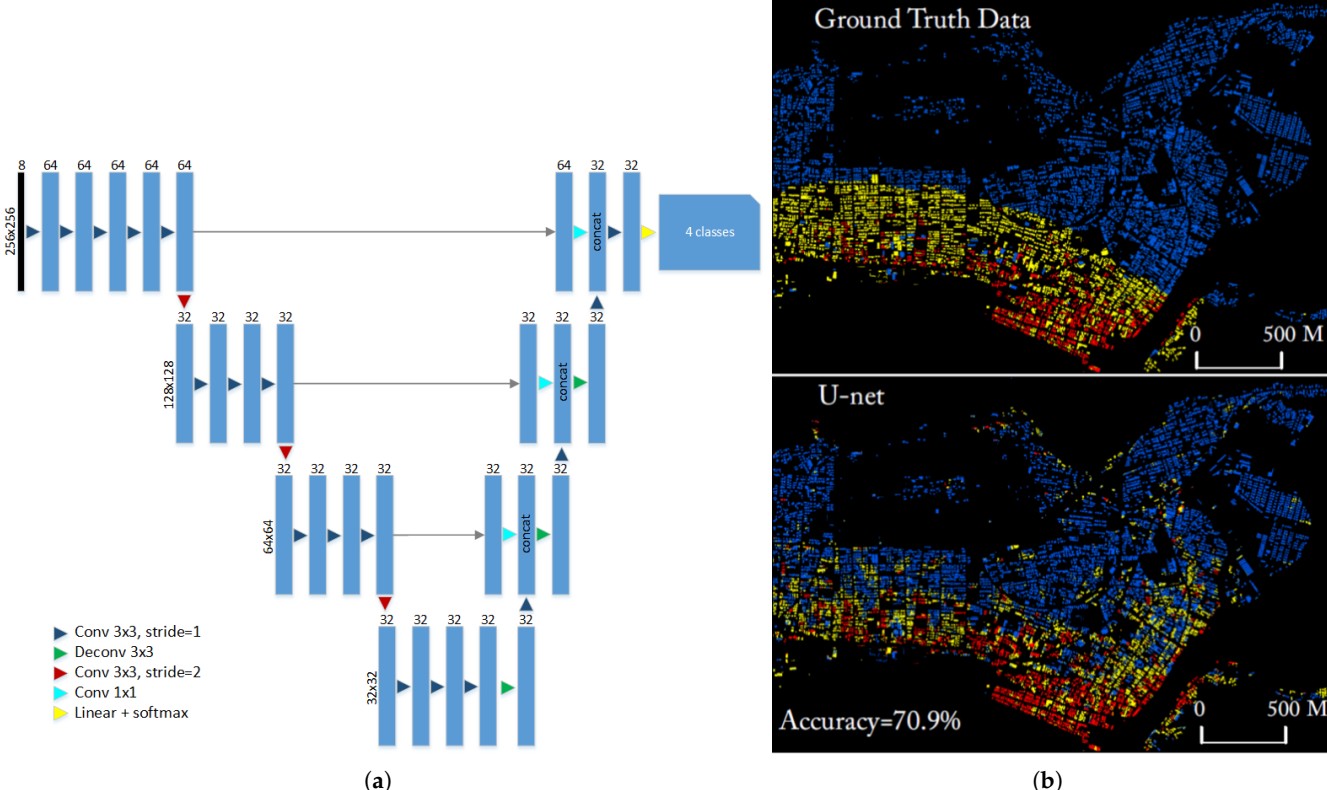

**Figure 11.** Application of the proposed modified U-Net by Bai et al. [17] on WorldView-2 VNIR images. (**a**) The modified U-Net architecture as described in their paper. Unless stated otherwise, all layers use batch normalization and Rectified linear Unit activation. (**b**) Ground truth of the damage and proposed U-Net damage assessment: washed-up buildings are in red, collapsed ones in yellow, intact ones in blue. Non-built areas are in black; The image is from [17] and was copied with the author's permission.

This modified algorithm is applied to concatenated WorldView-2 VNIR images of the Miyagi prefecture taken before and after the tsunami, for a total of eight channels. Three application areas are considered: Ishinomaki, Minamisanriku and Onagawa, and a three-class problem where buildings should be classified as washed away, collapsed or intact. A fourth class is considered for areas with no buildings. The authors achieved 70.9% overall accuracy with their algorithm, compared with only 54.8% for a Deep residual U-Net algorithm [74]. The visual results of their proposed method are shown in Figure 11b for the Ishinomaki area. As can be observed, the proposed classification is mostly consistent with the ground-truth. There is a sort of gradient, with washed-up buildings along the coastline, and the damage lessening inland, which the U-Net algorithm seems to have caught properly, despite a few misclassified buildings far inland. The most frequent misclassification appears to be between washed-away and collapsed buildings. One last comment that we can make from the visual results is that the authors appear to have projected their modified U-Net results within the segmentation provided by their ground-truth (zoom in on Figure 11b for details), and, as such, it is impossible to evaluate the segmentation part of the U-Net algorithm, which can also be a source of error and misclassification.

This paper is a good example of what collaboration between the fields of artificial intelligence and remote sensing should be. It is, in our opinion, a solid application with many good qualities: use of a state-of-the-art algorithm, and adaptation of said algorithm to the data, rather than adapting the data to an on-the-shelf algorithm. Regarding the weaknesses, besides the addition of batch normalization, the other modifications made to the original U-Nets are not always justified, resulting in a network that seems to work but looks odd. While some choices might have been made for speed or optimization purposes,

they are not clearly explained or justified from an AI perspective in the paper. Furthermore, the Deep residual U-Net with which the authors compared their algorithm seems to have been taken on-the-shelf and not tuned to the data, which may well explain the gap in performance between the two algorithms.

While it was designed for broader applications than the Tohoku tsunami, we can also mention another work from Bai et al. where the authors propose a pyramid-Pooling-Module-based Siamese network for damage assessment from various disasters, such as tornadoes, earthquakes, fires, floods and tsunamis [44]. The proposed architecture is also based on convolutional neural networks: its backbone consists of a residual network that uses pre- and post-disaster images. It is followed by a Squeeze network, and finally by the pyramid-pooling module. The originality of the architecture resides in the two parallel branches (hence the name Siamese network) in the first part of the network for the pre- and post-disaster images, which makes it possible to mainly use the pre-disaster images to detect the building locations and areas of interest.

Finally, in [31], Fujita et al. also explore the use of CNN applied to the detection of washed-away buildings. Unlike the other papers mentioned in this review, they use aerial images, which leads to higher resolution, but smaller areas of study. However, their framework is the most flexible of this study, since it can be used with either post-disaster images only, or with images from before and after the tsunami, depending on the image availability:

- In the case when only post-disaster images are available, they use a simple CNN with the following architecture: convolution-pooling-convolution-pooling-convolution-convolution, followed by two fully connected layers for class prediction;
- In the case where both pre- and post-disaster images are available, they use a Siamese network with the same basis but two convolutional entry branches before the fully connected layers;
- A second scenario is proposed when pre- and post-disaster images are available: using the regular non-Siamese network with a concatenated six-channel input image.

The main weakness of the paper is that the exact parameters of the layers are not mentioned in the paper: the authors only indicate that they used familiar networks such as AlexNet [69] and VGG [75] as basis.

The authors tested their three architectures on a dataset made of images taken in August 2000 for the pre-disaster images and 11th March 2011 for the post-disaster images. All architectures achieved scores between 93% and 96% accuracy in the detection of washed-away buildings. Given the important time-gap between the images before and after the disaster (11 years), the extremely high scores for only two classes, and the lack of detailed information on the neural networks used (the detailed architecture is not shown inside the paper), it is difficult to comment on the results of this paper and the re-usability of the proposed architecture.

### 3.3.2. Unsupervised Deep Learning

In [27], the authors propose a fully unsupervised framework based on autoencoders [76] and the Deep Embedded Clustering algorithm [77] to process images from before and after the tsunami in the Fukushima region, with the goal of first detecting changes caused by the disaster, and then sorting them into different categories. Autoencoders are unsupervised neural networks that are trained by trying to reproduce an identical output to the network input, with the goal of learning a high-quality latent representation of the data at the network bottleneck. The authors of [27] have used this principle to propose a joint fully convolutional autoencoder that tries to reconstruct the post-disaster image based on the pre-disaster image and vice-versa. By doing so, the network learns all seasonal changes, such as light changes and vegetation changes, but is unable to map and properly reconstruct areas damaged by the tsunami, thus making these tsunami-induced changes easy to detect using a simple Otsu thresholding [78] on the mean square error between the reconstructed image and the real image. Once the non-trivial changes potentially caused by the tsunami

or the earthquake have been detected, the authors apply Deep Embedded Clustering [77] to these areas to sort the damage into the following classes: flooded areas with stagnant water, severely damaged buildings, and other damage. The full architecture framework is shown in Figure 12. In their study, the authors compare their joint auto-encoder with Restricted Boltzman Machines [79,80] for the change detection parts, and several variants of the K-Means [60] algorithm are used as a comparison for the Deep Embedded Clustering algorithm for the damage-clustering part.

Using their method on ASTER VNIR images from 24 July 2010, 29 November 2010 and 19 March 2011, the authors achieve an overall accuracy of around 84% for the separation of trivial changes from tsunami-induced damage, and around 86% overall accuracy in the detection of damaged buildings and flooded areas within the detected areas of interest. Visual results are shown in Figure 13.

While this last study is interesting due to its fully unsupervised nature, exploiting the latest deep learning technologies, as well as its comparisons with several other unsupervised methods, it has several flaws: first, the data used by the authors have a relatively low resolution. Further, the authors are clearly not tsunami experts, as can be seen from the very rough classification of the damages they propose: while this may be due to the date and optical nature of the images, the *flooded area* class studied in this paper focuses only on stagnant water (which is relatively easy to detect) and discards temporarily flooded areas, which were numerous after the tsunami. As for the *damaged building* class, in addition to the absence of a damage degree scale, it does not mark the difference between buildings damaged by the earthquake and ones damaged by the subsequent tsunami.

While it is not clear if a distinction between the buildings damaged by the earthquake and the ones damaged by the tsunami can be made from the sky using optical images, this is certainly a weakness. This weakness may also explain the high number of damaged buildings detected relatively far inland, and why the results of this study appear to be fragile. The same statement can probably be made for the flooded areas with stagnant water, as there is probably no solution to an unsupervised algorithm based on change detection that would enable it sort out what was caused by the tsunami and what was not.

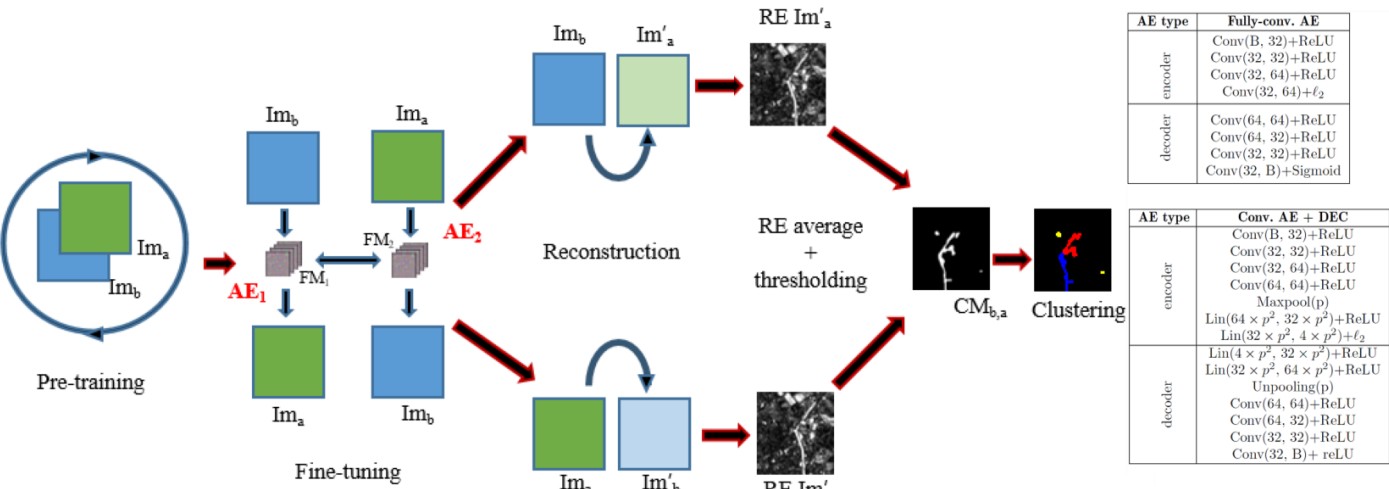

**Figure 12.** Unsupervised damage assessment architecture proposed by Sublime et al.: On the left, the different steps of the change detection and clustering process. On the right, tables detailing the architecture for their joint-autoencoder and for the Deep Embedding clustering Neural Network. The image is modified from [27].

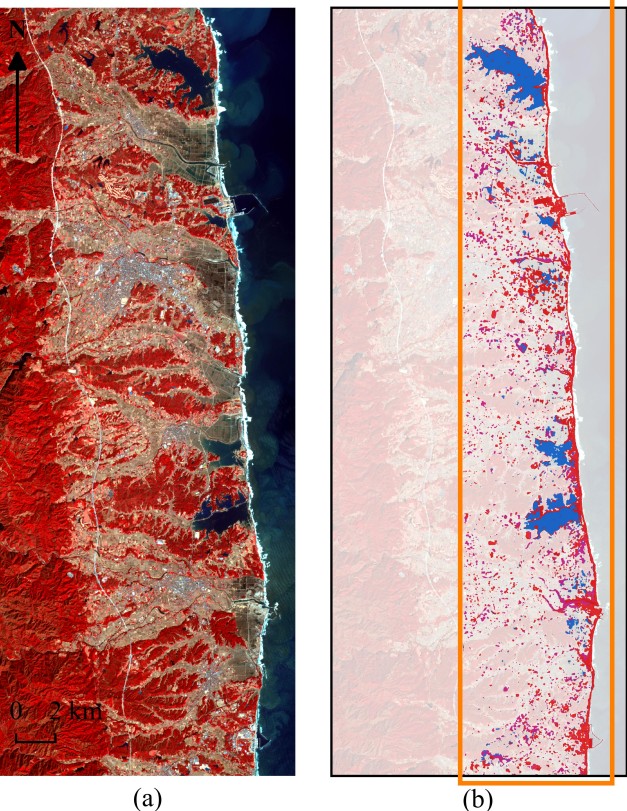

(a)  (b)

**Figure 13.** Results of Joint-Autoencoders and Deep Embedded clustering on ASTER VNIR images:
(**a**) Post-disaster image in false colors taken on 19 March 2011. (**b**) Damage assessment on the coastal
area sorted into 3 classes, flooded area in blue, damaged and destroyed buildings in red, and other
changes in purple. The image is from [27].

### 3.4. Results Comparison and Comments

In Table 3, we present a summary of the performances achieved by the various AI
methods discussed in this study. For each of them, we specify the type of source data,
the algorithm(s) used, the classes considered (the names might have been changed from
the original papers for homogeneity purposes), the metrics used, the score achieved, the
reference paper, and the year of publication.

The first thing that we can notice is that fully unsupervised methods which are,
in theory, the best choice for automated damage mapping form a minority of the used
methods. Furthermore, among the proposed unsupervised methods, some are not fully
unsupervised [14], some have not been qualitatively tested [26], and all of them focus on
classes that are a lot less specific than their supervised counterparts [14,26,27], with classes
which are sometimes so obscure that it is not clear what damage the study is actually
assessing [23]. We can also notice that these unsupervised and semi-supervised approaches
have surprisingly high scores. This can probably be explained by the less specific classes,
which allow these methods to remain competitive compared to supervised frameworks
that try to tackle more classes to describe the damage in more detail.

Another important thing that we can see from Table 3—and that we have already
mentioned in the metric section– is the type of metrics used to assess the quality of the
different methods. On the one hand, the use of different and non-overlapping metrics
makes comparisons difficult between the different papers, but, on the other hand, the lack
of diversity in the metrics, and the absence of some of them, is also a problem. While
many of the proposed studies provide either the F1 score or the Kappa index, some only
indicate the overall accuracy. This is problematic in the sense that the accuracy alone makes
it difficult to identify the strengths and weaknesses of a method. In particular, with only
the accuracy, it is impossible to detect high false-positive rates or high false-negative rates

in given classes of interest. Since these can be caused by either class imbalance or by a wrongly tuned algorithm, not having this information is a problem, especially given that the majority of these studies were conducted by non-AI experts, who could easily overlook tuning issues with an AI method that they do not know well.

**Table 3.** Chronological results of the different AI methods applied to images of the Tohoku tsunami for damage assessment or identification. OA is for Overall Accuracy, F1 for the F1 score, FPR for false positive rate, and FNR for false negative rate. See Figure 5 for the detailed class names. [1] The authors only mention a positive-change class, a negative-change class, and a no-change class, without more details. [2] block approach results. [3] La is for land areas, Fo for foam and Wa for water, either the sea or flooded areas. [4] Ignoring the 98% accuracy to detect majority class of non-built areas.

| Input | Method | Supervision | Classes | Metric | Scores | Ref. | Year |
|---|---|---|---|---|---|---|---|
| VNIR+TIR | KMeans | No | NC, SED, Wa, (+4) | - | - | [26] | 2013 |
| PolSAR | EM+MRF | No | +, −, 0 [1] | OA | 0.90 | [23] | 2013 |
| SAR | Decision Trees | Yes | WA, C, SLD | OA | 0.59–0.67 | [8] | 2015 |
| | | | | Kappa | 0.38–0.47 | [8] | 2015 |
| SAR | SVM | Yes | Change or not | F1 | 0.85 | [9] | 2016 |
| RGB | AlexNet+VGG | Yes | WA, C, NC | OA | 0.93–0.96 | [31] | 2017 |
| SAR | Regression+IHF | Semi | C, NC | F1 | 0.80–0.85 | [14] | 2018 |
| | | | | OA | 0.79–0.84 | [14] | 2018 |
| SAR | SVM | Yes | WA, MOD, SLD | F1 | 0.71–0.84 | [10] | 2018 |
| PolSAR | SVM | Yes | SED, MOD, SLD, I | OA | 0.89–0.95 [2] | [24,25] | 2018 |
| VNIR | UNet | Yes | WA, C, NC | OA | 0.71 | [17] | 2018 |
| | | | | F1 | 0.35–0.76 | [17] | 2018 |
| VNIR | DR-UNet [74] | Yes | WA, C, NC | OA | 0.55 | [17] | 2018 |
| | | | | F1 | 0.24–0.58 | [17] | 2018 |
| SAR | SqueezeNet+WRN | Yes | WA, C, SLD | OA | 0.71 | [11] | 2018 |
| SAR | SVM | Yes | NC, I, C | F1 | 0.80–0.91 | [12] | 2019 |
| VNIR | AE+DEC | No | C, NC, Wa | OA | 0.83–0.90 | [27] | 2019 |
| | | | | Kappa | 0.52–0.80 | [27] | 2019 |
| VNIR | AE+KMeans | No | C, NC, Wa | OA | 0.86–0.91 | [27] | 2019 |
| | | | | Kappa | 0.42–0.81 | [27] | 2019 |
| VNIR | SVM | Yes | La, Fo, Wa [3] | FPR | 0.59–0.83 | [16] | 2020 |
| | | | | FNR | 0.19–0.89 | [16] | 2020 |
| RGB | SSNet+PPM | Yes | WA, SED, SLD, I | OA | 0.69 [4] | [44] | 2020 |
| SAR | SVM + IHF | Semi | C, NC | OA | 0.85 | [15] | 2021 |
| | | | | F1 | 0.86 | [15] | 2021 |

Regarding the methods used, as we have mentioned in the previous paragraph, most of the regular Machine-Learning algorithms used in the studies presented in this review were fairly basic, and far from the non-deep-learning state-of-the-art: SVM was mostly used for supervised methods, and prototype-based approaches such as K-Means and EM for the unsupervised ones. However, the review has clearly shown a real interest from the Geoscience community in Deep Learning methods, and some real efforts to adapt these methods to their problems.

Finally, and since we have sorted the study in a chronological order, we can see that, surprisingly enough, the metric scores did not increase as much as expected over time, even when using deep learning approaches. This stagnation may have several causes. One of them could be that, despite the improvements in AI methods, there are inherent limitations in the source data from 2011t, which make it difficult to reach better results, even with modern Deep-Learning methods. Another possibility might be a too-heavy reliance on Deep-Learning methods, which led to the removal of expert-driven data pre-processing. Indeed, in the case of the simpler methods presented in this review, the pre-processing was a lot more consequent. It is, therefore, possible that convolutional neural networks do not yet have the ability to fully replace this expert knowledge. Finally, one last cause could be the complexity of Deep-Learning methods, which are quite difficult to tune properly [81] without strong collaborations between deep-learning experts and field experts.

## 4. Discussion and Conclusions

In this review paper, we have presented 10 years of artificial intelligence and more than 15 studies applied to the mapping of damage caused by the 2011 Tohoku tsunami. We have seen how the methods have evolved from the manual use of thresholds and feature combinations based on source remote-sensing data, to the use of Machine-Learning algorithms capable of classifying the damage, and finally the application of powerful Deep-Learning methods. The various studies presented in this work have shown much progress in this subject, and these methods have been successfully re-applied to other tsunamis, such as the 2018 Sulawesi earthquake and tsunami [82–85], and also to other types of disaster [44].

Yet, while there is room for optimism, we can clearly see that we are far from having achieved the full automation of damage-mapping using remote-sensing images and artificial intelligence. A large number of studies still heavily rely on manual data annotation, either relying directly on the source data, or to feed a supervised machine-learning algorithm. While it is true that some of these models could, in theory, be re-used for future geo-disasters, in practice, transfer learning remains a difficult problem in machine learning, and there is no guarantee that any model trained on annotated data from the Tohoku tsunami could be directly re-applied to another disaster. First, several remote-sensing programs have been stopped or evolved since 2011 (see Table 1), and any model—especially deep learning ones—trained on these data cannot be directly re-used on another data format with different resolutions or channels. Then, optical images (VNIR, PAN and RGB) can vary greatly from one country to another due to differences in the vegetation and, most importantly, in the local architecture of the buildings, thus making a model trained over one area difficult to reapply to a different country, even for images of the same format and resolution. Radar data might be less affected by this issue, but differences in building size or shape, as well as very different landscapes, could also make it difficult to transfer a model from one area to another. Attempts at generalization have been made with SVM, with fair results [9], but remained limited to the two-change and no-change classes on different areas of the same disaster, and such a generalization or transfer remains, in our opinion, impractical in the vast majority of cases, particularly with deep-learning models.

These issues can be solved in two ways. The first one would be to add an extra layer of artificial intelligence on top of the existing supervised methods. Neural networks such as Generative Adversarial Networks [86] are capable of transforming data from one format to another and even to upscale images [87], which could solve the issues of different source data. However, this extra layer also increases the risk of error propagation, and may hinder the quality of the final results. The other solution consists of increasing the number of unsupervised learning methods dedicated to tsunami damage assessment, while enriching them with expert knowledge to interpret their results and properly map the clusters to classes of interest. While the issue of evolving data formats remains true for these unsupervised algorithms, they have the huge advantage of not needing labeled data, and can thus be easily re-trained on new data with only minor modifications to account for the eventual changes in the number of channels or the resolution. A few of these unsupervised algorithms have been explored in studies on the 2011 Tohoku tsunami, but models such as W-Net [88] have not been tested, and could be a great unsupervised alternative to the U-Net study from Bai et al. [17]. W-Nets are unsupervised neural networks that first segment the input image, and then cluster the segments based on their similarity. Furthermore, the recent coupling of W-Nets with generative adversarial networks has already proven to be very useful for change detection from remote-sensing images [89]. Therefore, it seems to us that W-Net is a very good candidate for automated tsunami-damage assessment.

Finally, since unsupervised learning is known to have its own flaws—such as difficulties in mapping the clusters to classes of interest—the question of semi-supervised learning should not be ignored. While transfer learning remains difficult for complex deep-learning models, the transfer of simpler models, or the use of supervised expert knowledge within unsupervised frameworks, remains an interesting research lead. In this regard, the work of

Moya et al. on fragility functions and the demand parameter depending on the disaster intensity and partial field information seems to be promising [14,15].

To conclude, as we have mentioned throughout this review, we want to insist on the necessity of cross-discipline cooperation. Indeed, it is our opinion that difficult tasks such as automated damage assessment after a tsunami can only be successfully achieved if field experts from Artificial Intelligence, Geosciences, and Remote Sensing work together to tackle the different aspects of the problem.

**Funding:** This research received no external fundings.

**Acknowledgments:** This review would not have been possible without the tremendous amount of work done and the large number of papers published on the subject by Professor Shunichi Koshimura and his team; Several figures in this paper have been re-used from different journal papers under the fair use framework and with the authors' agreement, and I would like to thank all of them for their work.

**Conflicts of Interest:** The author declares no conflict of interest.

## Abbreviations

The following abbreviations are used in this manuscript:

| | |
|---|---|
| AE | Autoencoder |
| AI | Artificial Intelligence |
| ALOS | Advanced Land Observing Satellite |
| CNN | Convolutional Neural Networks |
| DEC | Deep Embedded Clustering |
| GLCM | Gray Level Co-occurrence Matrix |
| IHF | Imagery, Hazard, and Fragility function |
| JAXA | Japan Aerospace eXploration Agency |
| LiDAR | Light Detection And Ranging |
| MOD | Moderate Damages |
| NC | Not Collapsed |
| NDVI | Normalized Difference Vegetation Index |
| NOD | No Damages |
| OA | Overall Accuracy |
| PoLSAR | Polarimetric L-band Synthetic Aperture Radar |
| RGB | Red, Green, Blue |
| SAR | Synthetic Aperture Radar |
| SED | Serious(ly) Damage(d) |
| SLD | Slight(ly) Damage(d) |
| SVM | Support Vector Machine(s) |
| TIR | Thermal InfraRed |
| VNIR | Visible and near InfaRed |
| WA | Washed away |
| WRN | Wide Residual Network |

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
