# Peer review of "The 2011 Tohoku Tsunami from the Sky: A Review on the Evolution of Artificial Intelligence Methods for Damage Assessment"

_geosciences, doi:10.3390/geosciences11030133_

Round 1
Reviewer 1 Report
Improving methods for using artificial intelligence to assess damage from a tsunami is a challenging because of the complexity of the algorithms and it is a relatively new science. You have done a good job of summarizes the previous research. However, much of this is already presented in Koshimura et al. (2020). I suggest shortening the parts of the paper that have already been presented and direct readers to Koshimura et al's work.
Koshimura et al. (2020) does not extensively cover Deep Neural Networks (DNNs). I suggest you concentrate on DNNs, going into enough detail that a reader who is not familiar with DNNs can understand how they are formulated and used.
I also suggest you focus on the giving details on the metrics for comparing the performance of DNNs relative to other AI methods. In essence, much of the paper could be on Table 3 and include a detailed quantitative presentation of the different metrics used to evaluate AI.
Author Response
First, we want to thank the reviewer for the time they took reading our manuscript and for their very insightful comments.
We have tried our best to adress them as follows:
You have done a good job of summarizes the previous research. However, much of this is already presented in Koshimura et al. (2020). I suggest shortening the parts of the paper that have already been presented and direct readers to Koshimura et al's work.
We followed your advice and have shorten several sections regarding algorithms presented in Koshimura et al, and we have removed the less interesting contributions in terms of AI. Yet, we have kept some reasonable overlap with their review as we believe that a review should still cover as many authors as possible and because we needed to discuss some of these methods for metholodolical comparison purposes.
Furthermore, while this was already done in the original text, we have added further redirection to Koshimura et al.'s paper.
Koshimura et al. (2020) does not extensively cover Deep Neural Networks (DNNs). I suggest you concentrate on DNNs, going into enough detail that a reader who is not familiar with DNNs can understand how they are formulated and used.
This is an interesting idea, and we have improved the level of details given to the DNN sections. However, we feel like a review focusing only on deep learning methods might overlook too many contributions on the subject of tsunami damage assessment. Furthermore, the Koshimura review seems to have missed a few non-deep methods from other teams that we feel deserve to be mentioned.
Furthermore, due to the variety of neural networks used in the studies presented and the complexity of these algorithms, we are not convinced that a vulgarisation in 30 pages can bring much to the geoscience community. The details of these algorithms are in our opinion out of scope for this journal. As the other reviewer suggested, we want to keep the paper friendly and easy to read.
I also suggest you focus on the giving details on the metrics for comparing the performance of DNNs relative to other AI methods. In essence, much of the paper could be on Table 3 and include a detailed quantitative presentation of the different metrics used to evaluate AI.
Following your comment, we have added a more complex description of the different indexes in section 3.1, including their formulae, the core idea behind them, and their strengths and weaknesses. Thank you for this suggestion.
We have also do an additional round of proof-reading. We hope that you will find these changes satisfactory.
Reviewer 2 Report
This paper investigates the use of remote sensing in combination with Machine Learning techniques to identify and classify damage inflicted by the 2010 Tohoku event along the Sendai coast. In order to do that it provides a review of existing studies, analyzes and classifies the various Artificial Intelligence techniques by order of complexity and needed degree of human supervision and amount of data.
The authors conclude that current state of the art AI and ML technique still need a significant amount of human interaction and do not produce as reliable results those produced manually. This seems an honest assessment of the results they present.
There are two aspects of the material presented that I think require further attention by the authors:
-In the study results shown in Fig. 15 one of the features that get labeled is whether a particular area is inundated or not as a way of assessing damage to the buildings in it. However, the remote sensing images only show standing water after the tsunami has run up and down. There are extensive areas that were temporarily flooded during the wave run-up that are no longer flooded a few hours after wave arrival and do not appear in the images, however, the buildings in those areas may have suffered heavy damage from the wave. This is a point that seems to be overlooked in the study and that they authors should point out.
-Also the damage identified is attributed to the tsunami (line 532), however quite a lot of the damage to buildings in the tsunami inundation zone were caused by the earthquake and therefore may extend beyond flooded areas. More over, as stated earlier, the flooded areas shown in the images are most likely not the only ones inundated by the tsunami. Is there any way of differentiating those?
For the most part the paper seems well-documented and structured in a way that is easy to follow and understand by the general reader even if not an expert in ML. The assessment seems to be honest and realistic about the capabilities of existing AI techniques for damage evaluation after this type of natural disasters.
References 1,2,3, 4,6,7,... (I stopped checking there) appear on the reference list, but do not seem to be cited in the text. Please, make sure that only references mentioned in the text are included in the reference list.
The writing style and the command of the English language is good even though there are some grammatical errors having to do mostly with the use of the wrong verb form. I have indicated some of these problems with suggestions below, but not all of them. The authors should proof-read the manuscript for thus type of minor grammatical errors:
-line 15, I suggest "cause havoc" instead of "laid to waste"
-line 17 "10 years later" rather than "10 years after"
-line 103 "we will detail"
-line 217 "that only use images"
-line 227 "thus be considered"
-line 235 "two reason to use"
-line 304. Please define what an F1 score is.
-line 369 "to tune a Machine Learning algorithm instead of a long tedious manual map annotation"
-line 413: "clusters have to be merged"
-line 484 "in a network"
-line 524 "on this area" or "on these areas"
-line 528 "Deep Embedded Clustering algorithm"
-line 585 "brought much progress" rather than "brought many progresses"
Author Response
Firts, we want to thank our reviewer for the time spent on our paper and for the insightful feedbacks that were provided.
Below we give some answer to your comments and how we have adressed them:
This paper investigates the use of remote sensing in combination with Machine Learning techniques to identify and classify damage inflicted by the 2010 Tohoku event along the Sendai coast. In order to do that it provides a review of existing studies, analyzes and classifies the various Artificial Intelligence techniques by order of complexity and needed degree of human supervision and amount of data.
The authors conclude that current state of the art AI and ML technique still need a significant amount of human interaction and do not produce as reliable results those produced manually. This seems an honest assessment of the results they present.
Thank you very much for the kind comments.
In the study results shown in Fig. 15 one of the features that get labeled is whether a particular area is inundated or not as a way of assessing damage to the buildings in it. However, the remote sensing images only show standing water after the tsunami has run up and down. There are extensive areas that were temporarily flooded during the wave run-up that are no longer flooded a few hours after wave arrival and do not appear in the images, however, the buildings in those areas may have suffered heavy damage from the wave. This is a point that seems to be overlooked in the study and that they authors should point out.
-Also the damage identified is attributed to the tsunami (line 532), however quite a lot of the damage to buildings in the tsunami inundation zone were caused by the earthquake and therefore may extend beyond flooded areas. More over, as stated earlier, the flooded areas shown in the images are most likely not the only ones inundated by the tsunami. Is there any way of differentiating those?
Thank you for the very valid comments. We have added the comments you suggested in the main text concerning the detection of standing water only, and the challenges posed by detecting buildings damged by the earthquake and the ones damaged by the tsunami.
While we cannot rewrite the paper [26] from what is no Figure 13 (we would have liked to have these comments when writing it though ^^), we have adressed your concerns in the comments regarding this work in the main text : Given that this study uses an unsupervised algorithm and average resolution VNIR images, we do not think that it is possible to separate buildings damaged by the earthquake from the ones damaged by the tsunami (hence the focus on coastal areas). As for the issue of flooded areas, we fully agree with the limit you pointed out that it only detects standing water that remained after the tsunami. These elements have been mentioned in the text.
For the most part the paper seems well-documented and structured in a way that is easy to follow and understand by the general reader even if not an expert in ML. The assessment seems to be honest and realistic about the capabilities of existing AI techniques for damage evaluation after this type of natural disasters.
Thank you for the kind comment.
References 1,2,3, 4,6,7,... (I stopped checking there) appear on the reference list, but do not seem to be cited in the text. Please, make sure that only references mentioned in the text are included in the reference list.
We apologies, but we are not sure that we understand your comment. We use bibtex and so we are 100% certain that all references in the paper do appear in the manuscrit. We have recompiled several time just in case. We have also tried the following in case we did not understand what you meant :
- If you are refering to citations of form [6-14] to mention references 6 to 14, this is the MDPI template that compresses them in this form. We find it convenient to squeeze the references inside tables, but if you think it should not be done this way, we will definitely check with the editor for a solution.
- For reports references such as 2, 56 and 59 we had to put them into references since MDPI doesn't allow footnotes, but there is not much to say about them beyond citing them when we mention the numbers.
- If you were refering to the references in Table 2 : We have removed the "non-AI" references from Table 2 and moved them into the text. As for the other references of Table 2, we explain and detail all the algorithms mentioned here in section 3.
- If you were refering to the references Table 1 or Table 3, I personally find Tables linking the studies by categories useful (even if said studies are detailed only later or not at all). But if you don't think it is pertinent, we can remove the reference column on both Tables.
The writing style and the command of the English language is good even though there are some grammatical errors having to do mostly with the use of the wrong verb form. I have indicated some of these problems with suggestions below, but not all of them. The authors should proof-read the manuscript
We have fixed the typos you mentionned, and we have found a few others during our extra rounds of proof-reading. We hope that you will find these changes satisfactory.
Round 2
Reviewer 1 Report
The additions to the paper, in particular a more thorough description and evaluation of metrics, make it more useful.
There are still a few places where the English needs to be improved. This could easily be accomplished by having a native English speaker make the corrections.
Author Response
The additions to the paper, in particular a more thorough description and evaluation of metrics, make it more useful.
Thank you for your feedback.
There are still a few places where the English needs to be improved. This could easily be accomplished by having a native English speaker make the corrections.
Thank you for your remark. I asked 2 colleagues (including one native speaker) to help me track down the remaining typos. We managed to fix :
- A few verbs with missing "s" at the 3rd person
- A few other verbs with the wrong tense
- Some forgotten or needless plurals
- A typo in the legend of Figure 5
- A few sentences that were a bit long and that we decided to cut into smaller and easier to grasp pieces.
There was few of them left as you said, and ideally we hope that we have removed all of them.
Thank you again for your time and help.